

# PAME: plasmonic assay modeling environment

Adam Hughes, Zhaowen Liu and Mark E. Reeves

The George Washington University, Washington, D.C., USA

## ABSTRACT

Plasmonic assays are an important class of optical sensors that measure biomolecular interactions in real-time without the need for labeling agents, making them especially well-suited for clinical applications. Through the incorporation of nanoparticles and fiberoptics, these sensing systems have been successfully miniaturized and show great promise for *in-situ* probing and implantable devices, yet it remains challenging to derive meaningful, quantitative information from plasmonic responses. This is in part due to a lack of dedicated modeling tools, and therefore we introduce PAME, an open-source Python application for modeling plasmonic systems of bulk and nanoparticle-embedded metallic films. PAME combines aspects of thin-film solvers, nanomaterials and fiber-optics into an intuitive graphical interface. Some of PAME's features include a simulation mode, a database of hundreds of materials, and an object-oriented framework for designing complex nanomaterials, such as a gold nanoparticles encased in a protein shell. An overview of PAME's theory and design is presented, followed by example simulations of a fiberoptic refractometer, as well as protein binding to a multiplexed sensor composed of a mixed layer of gold and silver colloids. These results provide new insights into observed responses in reflectance biosensors.

## INTRODUCTION

Plasmonic sensors refer to a class of label-free detection platforms that utilize the optical properties of metals as a transduction mechanism to measure physical, chemical and biomolecular processes. These sensors have been utilized in immunology research (*Pei et al., 2010*; *Tang, Dong & Ren, 2010*), drug discovery (*Chen, Obinata & Izumi, 2010*; *Kraziński, Radecki & Radecka, 2011*), DNA mutations (*Litos et al., 2009*), and in many other novel applications. The conventional configuration of a plasmonic sensor is a thin layer of metal, most commonly gold or silver, deposited on a glass chip and illuminated from below at oblique incidence. This induces plasmon excitations along the surface of the film. This design has been successfully commercialized by Biacore[TM], and has been extended to include multilayer and mixed-alloy films (*Sharma & Gupta, 2006*; *Sharma & Gupta, 2007*), and films deposited on optical fibers. Over the same time period, gold and silver nanoparticles (AuNPs, AgNPs) gained attention for their potential in drug delivery (*Jong, 2008*; *Wilczewska et al., 2012*), and their intrinsic sensing properties, as

Corresponding author
Adam Hughes,
hugadams@gwmail.gwu.edu

each individual nanoparticle acts as a nanoscale transducer. Surface plasmons localized to roughly a 50 nm region around the colloid (*Malinsky et al., 2001*) are excited by light of any angle of incidence, and exhibit strong electromagnetic hotspots (*Barrow et al., 2012*; *Cheng et al., 2011*) with greater sensitivity to their local environment than bulk films. This is especially true for non-spherical nanoparticles like nanorods and nanostars (*Yin et al., 2006*; *Nikoobakht & El-sayed, 2003*; *Kessentini & Barchiesi, 2012*), and leads to great field-enhancements for Raman spectroscopy (*Freeman et al., 1995*; *Sau et al., 2010*). While free solutions of colloids alone can serve as sensors (*Jans et al., 2009*; *Tang, Dong & Ren, 2010*), they are easily destabilized by surface agents, and alterations in salinity and pH of their surrounding environment, resulting in particle aggregation (*Pease et al., 2010*; *Zakaria et al., 2013*). Nanoparticle monolayers engineered through vapor deposition (*Singh & Whitten, 2008*), lithography (*Haes et al., 2005*), or self-assembly onto organosilane linkers (*Nath & Chilkoti, 2002*; *Brown & Doorn, 2008*; *Fujiwara, Kasaya & Ogawa, 2009*) now commonly replace their bulk film counterparts, since they retain the enhanced sensitivity and flexible surface chemistry of the colloids, while being less prone[1] to aggregation.

Label-free measurements are indirect and prone to false-positives, as it can be difficult to distinguish specific binding events from non-specific binding, adsorption onto the sensor surface, and changes in the environment due to heating, convection and other processes. To mitigate these effects, plasmonic sensors undergo an extensive standardization process. First, they are calibrated to yield a linear response to bulk refractive index changes. Next, the sensor surface is modified[2] to be neutral, hydrophillic and sparsely covered with covalently deposited ligands. To measure ligand-analyte association and dissociation constants, $k_a, k_d$, solutions of varying concentration of analyte are washed over the surface, and the response is measured and interpreted within a protein interaction model (*Pollard, 2010*; *Chang et al., 2013*). Each of these steps impose challenges for plasmonic sensors utilizing nanoparticle-embedded films. Primarily, the sensor response becomes highly sensitive to the film topology (*Quinten, 2011*; *Lans, 2013*), which is difficult to interpret and optimize without modeling tools. Furthermore, measurements of $k_a$ and $k_d$ require varying analyte concentrations under identical surface conditions. Commercial systems often employ multichannel-sampling on a single surface (*Attana, 2014*), or multiplex multiple surfaces in a single run (*ForteBio, 2013*), while researchers mostly rely on the presumption of identical sensor surface topology, chemistry and experimental conditions. Without a quantitative model for sensor response to protein binding, it is impossible to estimate how many molecules are adsorbed on the sensor. This information is critical to validating binding models; for example, whether a measured response is too large to be described by a 1:1 monomeric reaction. Is the amount of deposited ligand likely to lead to avidity effects? In non-equilibrium applications such as monitoring the hormone levels of cells, the analyte concentration is unknown and only a single excretion event may occur; in such cases, the ability to translate optical responses directly into quantitative estimations of ligand and receptor through modeling is paramount.

[1] Chemically-deposited nanoparticles are slightly mobile in the film, and still tend to form dimers, trimers and higher order clusters under certain conditions (*Scarpettini & Bragas, 2010*; *Hughes et al., 2015*).

[2] For planar gold chips, Dextran provides an optimal coating; however, for nanoparticles, short-chain alkanethiols and polyethylene glycols are preferred due to their smaller size (*Malinsky et al., 2001*; *Mayer et al., 2008*).

Researchers modeling plasmonic systems may opt to use monolithic design tools like COMSOL$^{TM}$ Multiphysics and Lumerical$^{TM}$ Solutions. While such tools offer comprehensive photonic design environments, they are quite general in design and carry more overhead than is needed to model the basic fiber and chip geometries described in this paper. On the other hand, related open-source tools are too disjoint to be effectively integrated into a single workflow. For example, thin-film solvers are widely available (for a comprehensive list, see *Optenso, 2014*), and the MNPBEM Toolbox (*Hohenester & Trugler, 2012*) offers a Matlab$^{TM}$ interface for nanomaterial design. Yet, to design materials in MNPBEM and integrate them into film solvers requires a customized pipeline. Furthermore, simple geometric fill models for nanoparticle-protein binding are not available in these tools, even though these fill models have been successful (*Klebstov, 2004*; *Lopatynskyi et al., 2011*; *Tsai et al., 2011*) in describing AuNP-protein binding in free solution. With an abundance of parameters, ranging from the nanoscale to the macroscopic, characterizing a biosensor can quickly become intractable and a specialized solution is needed.

Herein, the Plasmonic Assay Modeling Environment (PAME) is introduced as an open-source Python application for modeling plasmonic biosensors. PAME is a fully graphical application that integrates aspects of material science (material modeling, effective medium theories, nanomaterials), thin-film design, fiberoptics, ellipsometry and spectroscopy, with the goal of providing a simple framework for designing, simulating and characterizing plasmonic biosensors. PAME helps to illuminate non-obvious relationships between sensor parameters and response. After an overview of its theory and design, several examples are presented. First, PAME is used to model the refractometric response of an AuNP-coated optical fiber to increasing concentrations of glycerin. It is shown that the response peaks at $\lambda_{max} \approx 485$ nm , a result supported by experiment, even though the nanoparticles absorb most strongly at $\lambda_{max} \approx 528$ nm. Next, PAME simulates protein binding events onto a mixed layer of gold and silver nanoparticles in a multiplexed fiber setup. Finally, a brief overview of PAME's requirements, performance and future development is presented. Additional examples in the form of IPython notebooks (*Perez & Granger, 2007*), as well as video tutorials, are available in the Supplemental Information.

## THEORY AND DESIGN

Many plasmonic sensors can be modeled as a multilayer stack of homogeneous materials, also referred to as films or dielectric slabs, arranged on a substrate so as to transduce interactions between light and the stack. The substrate represents a light guide such as a chip or an optical fiber. The transduced signal is some optical property of the multilayer, commonly transmittance or spectral reflectance, or in the case of ellipsometry, changes in the reflected light's polarization state (*Moirangthem, Chang & Wei, 2011*). Much of the diversity in plasmonic sensors is therefore due to design parameters rather than dissimilar physics, and has been described thoroughly by BD Gupta (*Sharma & Gupta, 2006*; *Sharma & Gupta, 2007*; *Gupta & Verma, 2009*; *Singh, Verma & Gupta, 2010*; *Mishra, Mishra & Gupta, 2015*). PAME was designed specifically to model these types of systems.

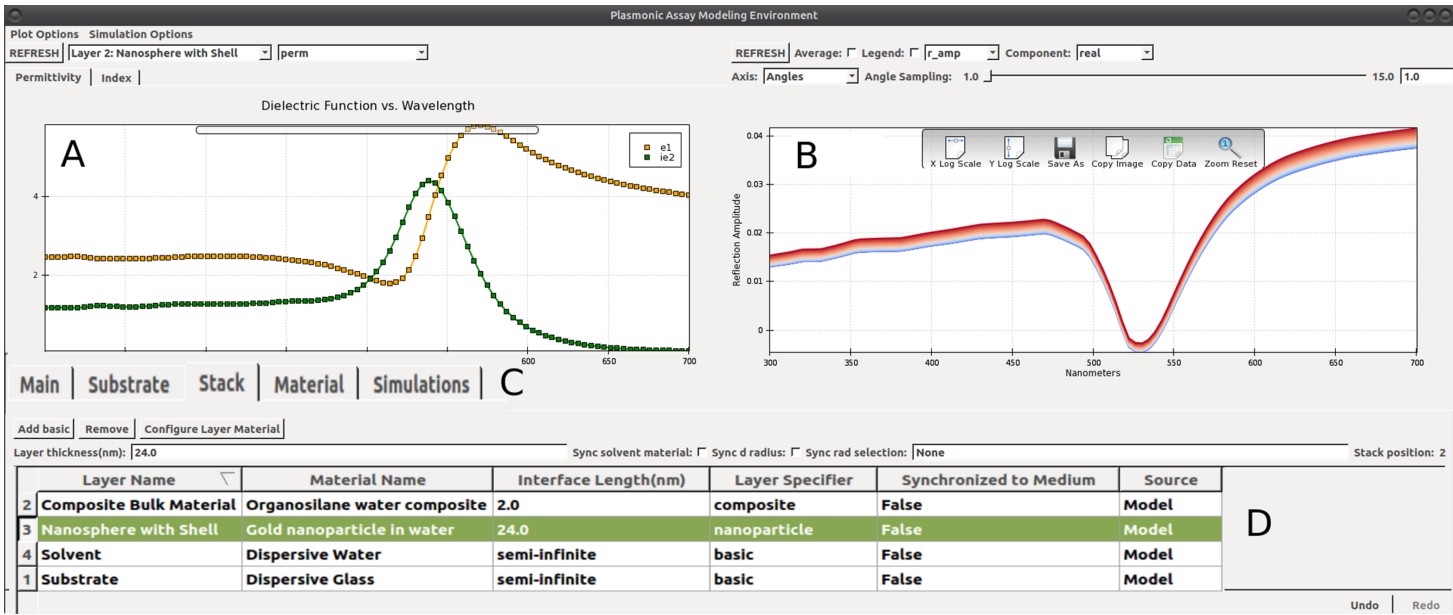

**Figure 1 PAME's user interface.** (A) Panel to view material quantities such as index of refraction, $n(\lambda)$, and nanoparticle extinction cross section, $\sigma_{ext}(\lambda)$. Currently shown is $e(\lambda)$ for a layer of gold nanoparticles in water at a fill fraction of about 30% using Garcia's mixing model. (B) Panel of plotted optical properties such as transmittance, $T(\lambda)$, and reflectance, $\Gamma(\lambda)$. Here the reflectance coefficient for P-polarized light $r_P(\lambda)$ is shown. The spread in the linewidth corresponds to variations in light modes over the range $0 > \theta \geq 16°$. (C) Five primary panels and tabular interface (D) for constructing the dielectric stack: silica (substrate) | organosilanes (2 nm) | AuNPs in $H_2O$ (24 nm) |$H_2O$ (solvent) for $300 \leq \lambda \leq 800$ nm.

PAME is designed with four integrated subprograms: a materials adapter to model bulk, composite, and nanomaterials; a multilayer thinfilm calculator; a substrate design interface; and a simulation and data analysis framework. Figure 1 shows a screenshot of the PAME's main window, with its five primary panels: **Main**, **Substrate**, **Stack**, **Material** and **Simulations**. Main refers to global settings, for example the operating wavelength range. The remaining tabs correspond directly to the four aforementioned subprograms. Together, they provide a complete framework for modeling a plasmonic sensor, and lend a useful narrative that will be followed in the ordering of the remaining sections of this paper. Incidentally, the progression from Substrate, Stack and Material represents a top-down view of the model, starting from macroscopic parameters and working down to the microscopic.

## Substrate types

PAME supports two substrates: optical fibers and chips. Substrates mediate the interaction between light and the multilayer stack through a weighting function, $\sum_i^N f(\theta_i)$, where $\theta_i$ corresponds to the angle of the $i$th incident light ray onto the substrate. The chip is meant to describe simple configurations, for example a gold film deposited on a glass slide and illuminated from below at a single angle, $\theta_o$. In this case, $\sum_i^N f(\theta_i) = \delta(\theta_o)$. For optical fibers, the propagation modes are determined by properties of the fiber itself, such as its numerical aperture, core and cladding materials, and its ability to maintain polarization states. Furthermore, the placement of the multilayer on different regions of the fiber has a

**Figure 2 Light propagation in fiber optic biosensor.** (A) A light ray propagating in an optical fiber core. Transverse refers to a multilayer deposited along the propagation direction, while axial is perpendicular and deposited on the fiber endface. The fiber cladding and jacket are hidden for clarity. (B) The $\theta = 0$ plane wave incident on the stack. (C) Illustration of the homogenized multilayers, and some of the electromagnetic quantities associated with each interface, reproduced with permission from (*Orfanidis, 2008*).

[3] See *Mishra, Mishra & Gupta (2015)* Eq. (5) for an example of $f(\theta)$ for a transversal fiber with collimated light source.

significant effect on $f(\theta_i)$, and hence on the optical response of the sensor. The two most common orientations, either transversally[3] along the propagation direction on the fiber, or axially on the cleaved fiber endface, are shown in Fig. 2. Both of these orientations have been realized as biosensors (*Lipoprotein et al., 2012*; *Shrivastav, Mishra & Gupta, 2015*), with the axial configuration, often referred to as a "dip sensor" because the endface is dipped into the sample, appearing more often in recent years (*Mitsui, Handa & Kajikawa, 2004*; *Wan et al., 2010*; *Jeong et al., 2012*; *Sciacca & Monro, 2014*). PAME does not presently support multilayers along bent regions of the fiber, or along sharpened optical tips (*Issa & Guckenberger, 2006*; *Library et al., 2013*) and assumes all rays to be plane waves. For advanced waveguide design and modal analysis, we recommend Lumerical™ MODE solutions.

PAME's **Substrate** interface queries users to configure chip and optical fiber parameters, rather than working directly with $f(\theta_i)$. Users can choose between multilayer orientation, polarization state (P, S or unpolarized), and range of angles, all from which PAME builds $f(\theta)$. The interface is user-friendly, and attempts to obviate incompatible or unphysical settings. For instance, the ellipsometric amplitude ($\Psi$) and phase ($\delta$) depend on ratios of P-polarized to S-polarized light reflectance, but users may opt to only compute S-waves, resulting in errant calculations downstream. Anticipating this, PAME provides an ellipsometry mode, which when enabled, prevents the polarization state from being changed. By combining substrate types with context modes, PAME provides a simple interface for modeling a number of common optical setups.

## Multilayer stack

Figure 2 depicts the multilayer stack, in which each dielectric slab is assumed to be homogenous, and of uniform thickness; heterogeneous materials must be homogenized through an effective medium theory (EMT). Furthermore, the multilayer model presumes that layers are connected by smooth and abrupt boundaries to satisfy Fresnel's equations. The first and last layers, conventionally referred to as "substrate" and "solvent," are assumed to be semi-infinite, with incident light originating in the substrate. The treatment

of anisotropic layers without effective medium approximations is discussed in the **Future Improvement** section.

A light ray incident on the stack at angle $\theta$, as set by $f(\theta)$, will reflect, refract and absorb, in accordance with Fresnel's equations. For example, in a simple 2-layer system, the light reflectance, $\Gamma(\lambda)$, at the boundary between $n_1, n_2$ is

$$R = \frac{1}{2}(r_s + r_p) \tag{1}$$

$$R = \frac{1}{2}\left( \left| \frac{n_1\cos\theta_i - n_2\cos\theta_t}{n_1\cos\theta_i + n_2\cos\theta_t} \right|^2 + \left| \frac{n_1\cos\theta_t - n_2\cos\theta_i}{n_1\cos\theta_t + n_2\cos\theta_i} \right|^2 \right), \tag{2}$$

where $\theta_i, \theta_t$ are the angles of incidence upon, and transmission into $n_2$ from $n_1$, and $r_s$ and $r_p$ are the complex reflection coefficients of the S and P-polarized light. For $N$-layers, Fresnel's equations are solved recursively using the transfer matrix method(TMM), also referred to as the recursive Rouard method (*Rouard, 1937*; *Lecaruyer et al., 2006*). In addition to the reflectance, transmittance and absorbance, a variety of optical quantities are computed in the multilayer, including the Poynting vector, the complex wave vector angle, ellipsometric parameters, and film color. For a thorough treatment of light propagation in multilayer structures, see *Orfanidis (2008)* and *Steed (2013)*. PAME offers a simple tabular interface for adding, removing, and editing materials in arbitrarily many layers, as shown in Fig. 2C. PAME delegates the actual TMM calculation to an adapted version of the Python package, tmm (*Byrnes, 2012*).

## PAME material classes

PAME includes three material categories: bulk materials, composite materials and nanomaterials. A bulk material such as a gold film is sufficiently characterized by its index of refraction. The optical properties of a gold nanoparticle, however, depend on the index of gold, particle size, the surrounding medium, a particle-medium mixing model, and other parameters. A nanoparticle with a shell is even more intricate. PAME encapsulates a rich hierarchy of materials in an object-oriented framework to ensure compatibility with the multilayer stack and interactive plotting interface.

### *Bulk material*

In PAME, a "bulk" material refers to a single, homogeneous substance, fully characterized by its complex index of refraction, $\tilde{n} = n + i\kappa$, or dielectric function, $\tilde{e} = e + i\epsilon$, which are related through a complex root ($\tilde{e} = \sqrt{\tilde{n}}$), which gives the relations

$$e = n^2 - \kappa^2 \qquad n = \sqrt{\frac{\sqrt{e^2 + \epsilon^2} + e}{2}}$$

$$\epsilon = 2n\kappa \qquad \kappa = \sqrt{\frac{\sqrt{e^2 + \epsilon^2} - e}{2}}.$$

Here $n$ and $e$, and optical quantities derived from them, are understood to be dispersive functions of wavelengths, $n(\lambda), e(\lambda)$. The refractive index $n$ is assumed to be independent

[4] Ferromagnetic nanoparticles do exist, and are have already been utilized in sensing applications (*Pellegrini & Mattei, 2014*).

of temperature and non-magnetic at optical frequencies[4]. The index of refraction of bulk materials is obtained through experimental measurements, modeling, or through a combination of the two; for example, measuring $n$ at several wavelengths, and fitting to a dispersion model such as the Sellmeier equation,

$$n(\lambda) = \sqrt{1 + \frac{A_1\lambda^2}{\lambda^2 - B_1^2} + \frac{A_2\lambda^2}{\lambda^2 - B_2^2} + \frac{A_3\lambda^2}{\lambda^2 - B_3^2} + \cdots}. \tag{3}$$

[5] Materials are supplied as is with no guarantee of accuracy: use at your own discretion.

PAME is bundled with several dispersion models, including the Cauchy, Drude and Sellmeier relations, as well as two freely-available[5] refractive index catalogs: *Sopra (2008)* and RefractiveIndex.INFO (*Polyanskiy, 2015*), comprising over 1,600 refractive index files. PAME includes a materials adapter to browse and upload materials as shown in Fig. 3. Selected materials are automatically converted, interpolated, and expressed in the working spectral unit (nm, eV, cm, ...) and range. PAME's plots respond to changes in material parameters in real time.

### Composite materials

A composite consists of two materials bound by a mixing function. For example, a gold-silver alloy could be modeled as bulk gold and silver, mixed through an effective mixing theory (EMT). The complexity of the EMT is related to the electromagnetic interactions between the materials. For example, for binary liquid mixtures with refractive indicies, $n_1, n_2$, and fill fraction, $\phi$, the composite can be approximated as $n_{\mathrm{mixed}} = n_1\phi + n_2(1 - \phi)$, with more complex liquid mixing models like Weiner's relation and Heller's relation yielding negligible differences (*Bhatia, Tripathi & Dubey, 2002*). For solid inclusions, the extension of the Maxwell–Garnett (MG) mixing rule (*Garnett, 1904*) by *Garcıa, Llopis & Paje (1999)* has been shown effective, even when the particles are non-spherical and anisotropically clustered (*Li et al., 2006*). At present, PAME includes MG, with and without Garcia's extension, the Bruggeman equation (*Bruggeman, 1935*), the quasi-crystalline approximation with coherent potential (*Liu et al., 2011*; *Tsang, Kong & Shin, 1985*), and various binary liquid mixing rules. These are hardly exhaustive, and new methods are continually appearing (*Amendola & Meneghetti, 2009*; *Battie et al., 2014*; *Malasi, Kalyanaraman & Garcia, 2014*). Adding EMTs to PAME is straightforward, and more will be added in upcoming releases.

Composite materials are not limited to bulk materials, but include combinations of composites and/or nanomaterials, for example gold and silver nanoparticles embedded in a glass matrix. However one must be aware of the limitations of implicit mixing models. For example, consider a layer of gold nanoparticles. As coverage increases, particle–particle interactions are taken into account in Garcia's EMT through the parameter $K$. EMTs describing inclusions of two or more material types have been described (*Zhdanov, 2008*; *Bossa et al., 2014*), and will be available in future versions. PAME's geometric fill models are also implemented as composite material classes. For example, small spheres of material X binding to the surface of a larger sphere of material Y serves as a useful model for proteins binding to gold nanoparticles (*Lopatynskyi et al., 2011*). An ensemble of spherical

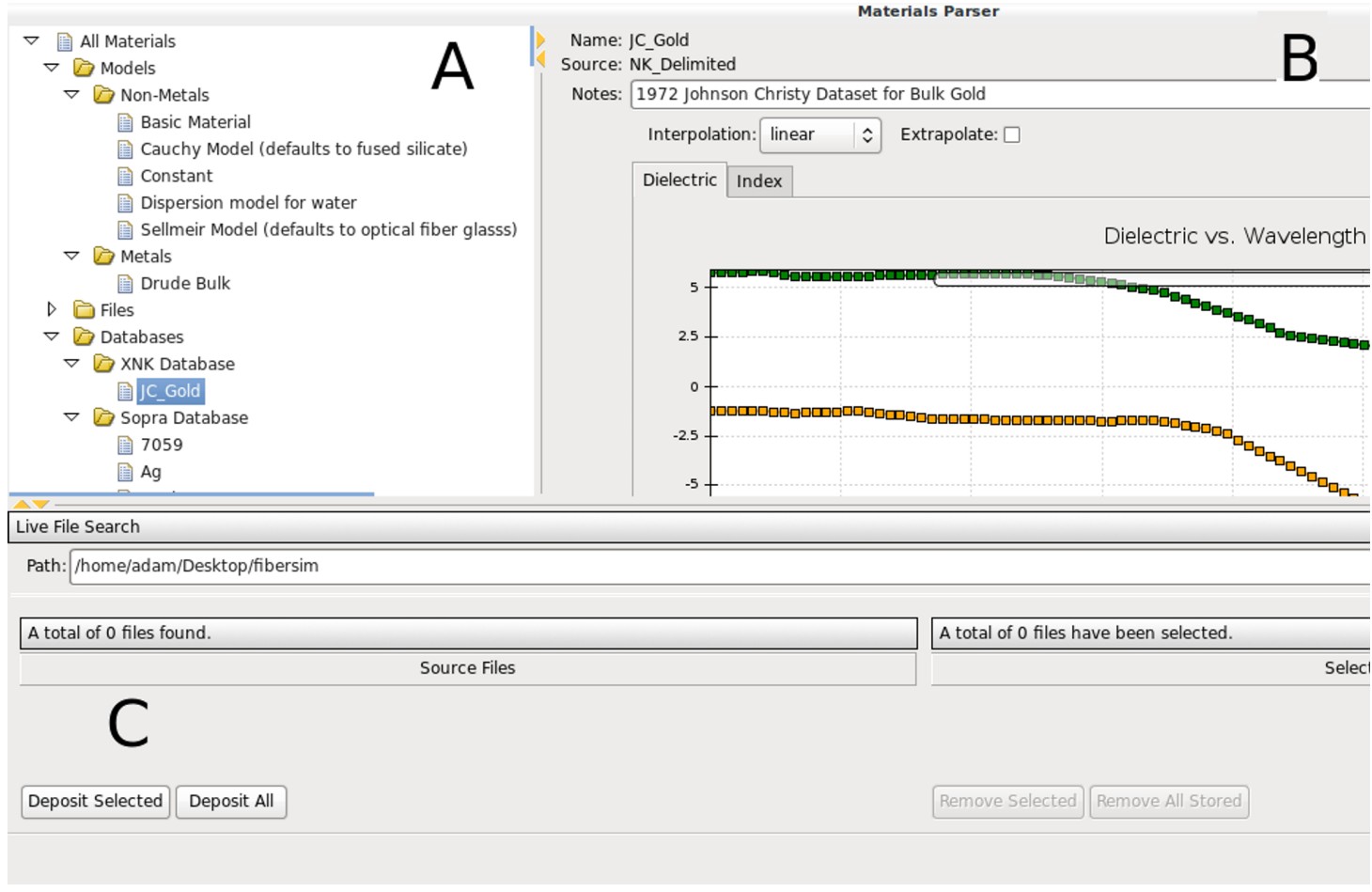

**Figure 3** **Screenshot of PAME's material adapter.** (A) Tree view of available bulk material models, files and bundled catalogues. (B) Preview of the selected material includes available material metadata, notes and an interpolated fit to the current spectral range and unit system. Currently showing $\tilde{\varepsilon}$ for gold from *Johnson & Christy (1972)*. (C) Search utility to find and batch upload materials.

inclusions on a disk is the correct geometry for modeling gold nanoparticles adhered to the cleaved end of a fiber surface. PAME's fill models track the number of inclusions, fill fraction, and other quantitative parameters at any given time. This enables macroscopic quantities like sensor sensitivity to be measured against microscopic parameters like the number of proteins bound to the nanoparticles.

### Nanomaterials

In PAME, nanomaterials are treated as a special instance of a composite material[6] whose properties depend on a core material, a medium material, possibly an intermediate shell material, and particle size. A key distinction between nanomaterials and their bulk counterparts is that the optical properties of nanoparticles are highly sensitive to both the particle size and the permitivity of the surrounding medium. The implicit optical properties of spheroidal nanoparticles, such a extinction cross section, $\sigma_{ext}$, are solved analytically through Mie Theory (*Bohren, 1983*; *Jain et al., 2006*). This is a fundamentally important quantity, as the position and shift in the extinction cross section maximum,

[6] A layer of nanoparticles is always embedded in some other media, for example in a slab of water or a sol–gel matrix of glass. Therefore, in an object-oriented framework, a nanomaterial is a subclass of a composite material, with additional attributes like particle size and implicit optical properties.

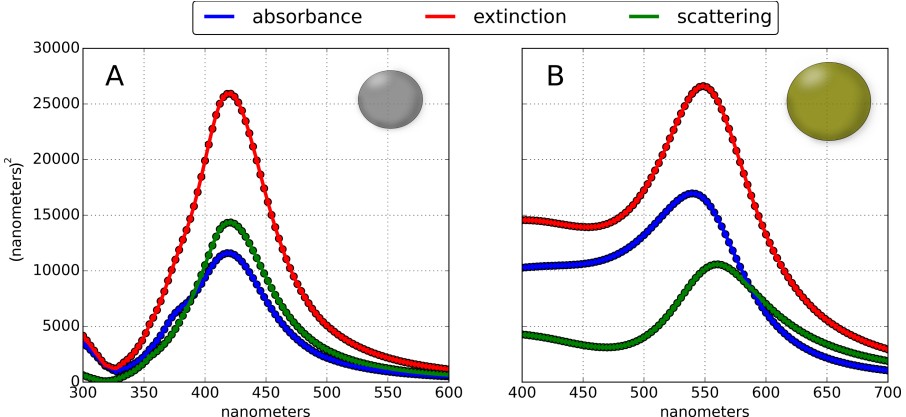

**Figure 4 Theoretical optical absorption and scattering properties of gold and silver nanoparticles.** Extinction, absorption and scattering cross sections of (A) 60 nm silver and (B) 80 nm gold nanoparticles computed in PAME using reported permittivities from *Hagemann, Gudat & Kunz (1975)* and *Gao, Lemarchand & Lequime (2011)*, respectively, which produced more accurate cross sections than the typically-used Drude model.

known as the localized plasmon resonance (*Willets & Van Duyne, 2007*; *Anker et al., 2008*), is often the best indicator of the state of the nanoparticles comprising the system. While the full solution to the extinction cross section is described by the sum of an infinite series of Ricatti–Bessel functions, described in full in *Lopatynskyi et al. (2011)*, for brevity consider the approximate expression (*Jeong et al., 2012*; *Van de Hulst, 1981*):

$$\sigma_{\text{ext}} \approx \underbrace{\frac{128\pi^5}{3\lambda^4}R^6 Im\left[\frac{m^2-1}{m^2+2}\right]^2}_{\approx \sigma_{\text{scatt}}} - \underbrace{\frac{8\pi^2}{\lambda}R^3\left[\frac{m^2-1}{m^2+2}\right]}_{\approx \sigma_{\text{abs}}}, \tag{4}$$

where $m$ is the ratio of the refractive index of the core particle material to that of the suspension medium (e.g., gold to water). The polynomial dependence of $R$, $\lambda$ and $m$ demonstrate the high variability in the optical properties of nanoparticles, and by extension, the biosensors that utilize them. Typical cross sections from silver and gold nanospheres are depicted in Fig. 4.

While optical constants derived from Mie Theory are computed analytically, it is important to recognize that in a dielectric slab, nanomaterials are represented by an effective dielectric function; thus, optical constants like transmittance or reflectance will be computed using an effective dielectric function representing the nanoparticle layer. PAME currently supports nanospheres and nanospheres with shells; planned support for exotic particle morphologies is described in the **Future Improvements** section. Similar treatments of nanoparticle layers with effective media approximations have been successful (*Li et al., 2006*; *Liu et al., 2011*), even for non-spherical particles, and for ensembles of different sized particles (*Battie et al., 2014*). This is a salient difference between PAME, and numerical approaches like the boundary element method(BEM), discrete dipole approximation(DDA) and finite-difference time-domain(FDTD): PAME relies on mixing theories, and hence is constrained by any underlying assumptions of the mixing model. For

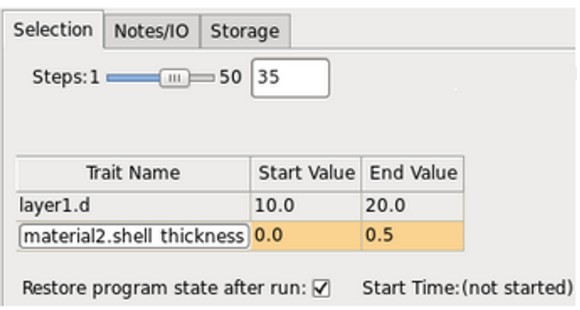

**Figure 5** PAME's simulation interface, showing the Selection, Notes/IO and Storage tabs.

a more in-depth discussion on nanoparticle modeling, see *Myroshnychenko et al. (2008)* and *Trügler (2011)*.

## Simulation and data analysis

PAME's interactivity makes it ideal for exploring the relationships between system variables, while the simulation environment provides the means to systematically increment a parameter and record the correspond response; for example, incrementing the fill fraction of inclusions in a nanoparticle shell to simulate protein binding, or incrementing the refractive index of the solvent to simulate a refractometer. Because most updates in PAME are automatically triggered, simulations amount to incrementing parameters in a loop and storing and plotting the results.

PAME's simulation interface simplifies the process of setting simulation variables and storing results. It is comprised of three tabs: a **Selection** tab (Fig. 5) for setting simulation parameters and value ranges. Variable names like *layer1.d* refers to the thickness of the first layer in the multilayer stack, and *material2.shell_thickness* is the size of the nanoparticle shell in nanometers of material2. PAME provides suggestions, documentation and a tree viewer to choose simulation variables, and alerts users to errant inputs, or invalid ranges; for example, if users try to simulate a volume fraction beyond its valid range of 0.0 to 1.0. The **Notes/IO** tab provides a place to record notes on the simulation, and configure the output directory. PAME can store all of its state variables in every cycle of the simulation, including the entire multilayer structure, and all computed optical quantities, but this can lead to storing large quantities of redundant data. The **Storage** tab lets users pick and choose their storage preference, and even specify the quantities that should be regarded as "primary" for easy access when parsing.

PAME provides a SimParser object to interact with saved simulations, which while not required, is intended to be used inside an IPython Notebook (*Perez & Granger, 2007*) environment. The SimParser stores primary results in Pandas (*McKinney & Millman, 2010*) and scikit-spectra (*Hughes & Liu, 2014*) objects for easy interaction and visualization, and the remaining results are stored in JSON. This allows for immediate analysis of the most important simulation results, with the remaining data easily accessed later through a tree viewer and other SimParser utilities.

# EXAMPLES OF USE

## Case 1: refractometer

Plasmonic sensors respond to changes in their surrounding dielectric environments, and are commonly utilized as refractometers (*Mitsui, Handa & Kajikawa, 2004*; *Punjabi & Mukherji, 2014*), even going so far as to measure the refractive index of a single fibroblast cell (*Lee et al., 2008*). Refractive index measurements can also be used to measure sensitivity and linear operating ranges. A common approach is to immerse the sensor in a medium such as water, and incrementally change the index of refraction of the medium by mixing in glycerine or sucrose. Because the index of refraction as a function of glycerine concentration is well-known (*Glycerine Producers' Association , 1963*), sensor response can be expressed in refractive index units (RIU). This is usually taken a step further in biosensor designs, where the RIUs are calibrated to underlying biophysical processes (e.g., protein absorption), either through modeling as PAME does, or through orthogonal experimental techniques such as Fourier Transform Infrared Spectroscopy (*Tsai et al., 2011*). This calibration process has been described previously (*Jeong & Lee, 2011*; *Richard & Anna, 2008*), and is usually carried through in commercial plasmonic sensors. This quantifies the analyte binding capacity of the sensor, an important parameter for assessing binding models[7], non-equilibrium sensing, and performing one-step measurements, for example estimating the glucose levels in a blood sample. As a first use case, PAME is used to calibrate sensor response to increasing concentrations of glycerine for an axial fiber comprised of a 24 nm layer of gold nanoparticles.

A dip sensor was constructed using a protocol and optical setup similar to that of *Mitsui, Handa & Kajikawa (2004)*. In brief, optical fiber probes were cleaved, submerged in boiling piranha (3:1 $H_2SO_4$ : $H_2O_2$), functionalized with 0.001% (3-Aminopropyl) trimethoxysilane for 60 min in anhydrous ethanol under sonication, dried in an oven at 120 °C, and coated with 24 nm AuNPs to a coverage of about $40 \pm 5$%, as verified by SEM imaging (*Hughes et al., 2015*). The fiber was submerged in 2 mL of distilled water under constant stirring, and glycerine droplets were added incrementally until the final glycerine concentration was 32%, with each drop resulting in a stepwise increase in the reflectance as shown in Figs. 6A and 6C). This system was simulated using the stack described in Fig. 1, where the organosilanes were modeled as a 2 nm-thick layer of a Sellmeier material (Eq. (4)), with coefficients: $A_1 = 6.9, A_2 = 3.2, A_3 = 0.89, B_1 = 1.6, B_2 = 0.0, B_3 = 50.0$. These coefficients led to excellent agreement between experiment and simulation during the self-assembly process of the AuNP film.

Figures 6C and 6D shows the strong agreement between measured and simulated response to increasing glycerine, and PAME is able to show the reflectance spectrum free of the influence of the LED light source in the dataset(b,a). It is clear that while the nanoparticle's reflectance is prominent around $\lambda_{max} \approx 560$ nm, the combination of both an increase in reflectance, and a blue-shift of spectral weight yield a 485 nm peak in the normalized reflectance spectrum. Neither is indicative of the free-solution plasmon resonance peak at $\lambda_{max} \approx 528$ nm , and maintaining a correspondence between the reflectance centroid and plasmon resonance can lead to misinterpretation. Furthermore,

[7] *Schasfoort et al. (2012)* has enumerated seven interfering effects that lead to errant calculations of equilibrium affinity constants. Estimations of nanoparticle and ligand density at the sensor surface provide insights as to whether or not some of these effects are likely occurring.

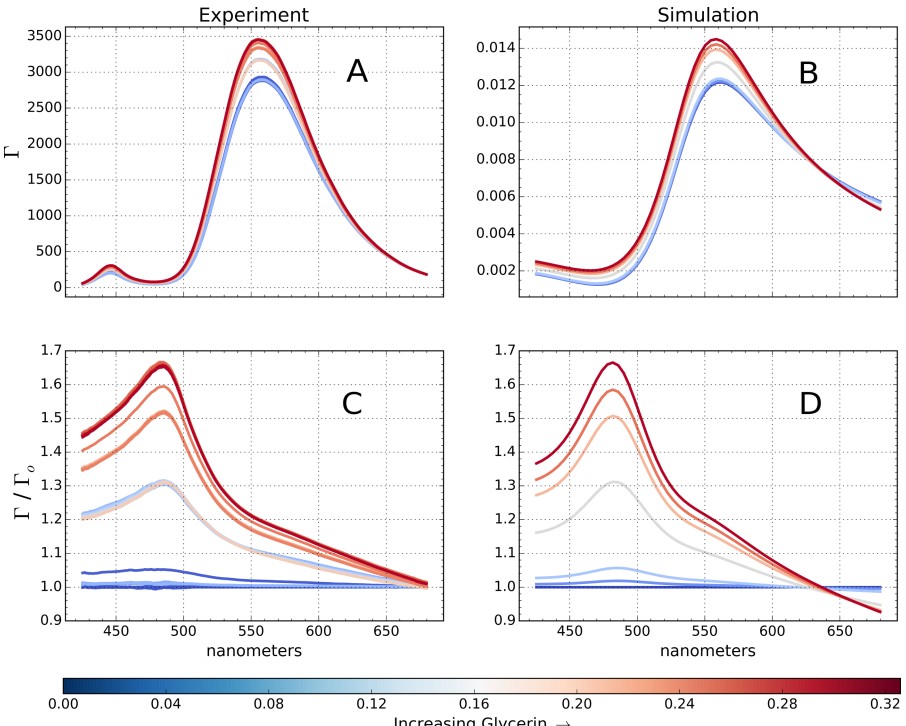

**Figure 6 Simulated and measured spectral reflectance from a fiberoptic biosensor.** Increase in spectral reflectance ($\Gamma$) from a fiber dip sensor at 40% AuNP coverage immersed in water glycerine mixture as glycerine fraction is increased from 0 to 32% for the experiment (A) and simulation (B). The same response, normalized to the reflectance of the probe in water ($\Gamma_o$) prior to the addition of glycerine (C, D). The AuNPs layer reflects most strongly at $\lambda_{max} \approx 560$ nm, yet the normalized reflectance peaks at $\lambda_{max} \approx 485$ nm.

the shape of this glycerine response profile is very sensitive to parameters like organosilane layer thickness and nanoparticle size and coverage, and by fitting to the simulated response, one may then estimate these parameters which are otherwise difficult to measure. This simple example provides valuable insights into the relationship between glycerine concentration and reflectance on a dip sensor.

## Case 2: multiplexed Ag–Au sensor

*Sciacca & Monro (2014)* recently published a multiplexed biosensor in which both gold and silver nanoparticles were deposited on the endface of a dip sensor and their reflectance was monitored simultaneously. In their experiment, the gold colloids were functionalized with anti-apoE, the antibody to an overexpressed gastric cancer biomarker, apoE. The silver colloids were functionalized with a non-specific antibody. The authors showed that the plasmon resonance peak of the gold particles shifted appreciably in response to apoE, while the silver did not. Furthermore, the gold peak did not respond to CLU, an underexpressed gastric cancer biomarker while the uncoated silver particles did, presumably due to non-specific binding. In effect, the multiplexed sensor provides a built-in negative control and can identify specific events more robustly. The ability to multiplex two or more colloids to a single sensor has great potential. To gain insights into

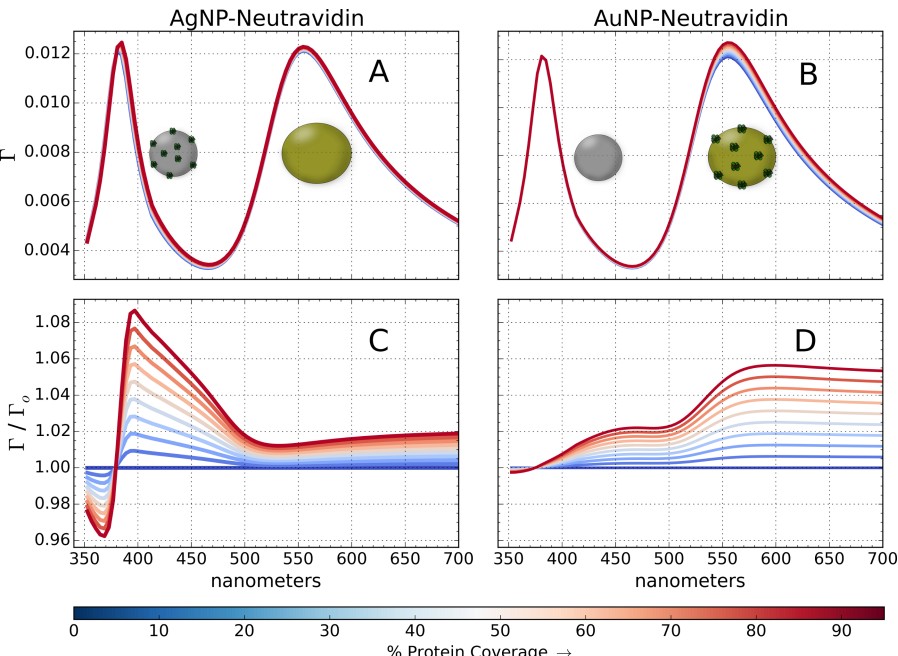

**Figure 7** **Simulation of multiplexed biosensor response.** Simulation of a multiplexed biosensor with a combined 80 nm gold and 60 nm silver nanoparticle layer. Simulated NeutrAvidin binding to AgNPs (A, C) while AuNPs are kept bare, and vice versa (B, D). The coverage is varied until 95% of the available sites are occupied (617 proteins per AuNP, 363 per AgNP). The reflectance normalized to zero coverage ($\Gamma_o$) is shown in (C, D).

[8] The NeutrAvidin simulation is an idealization of *Sciacca & Monro (2014)*'s configuration, as it only considers a single protein layer rather than an antigen-antibody bilayer.

[9] *Sciacca & Monro (2014)* actually used a thick layer of PAH to bind the nanoparticles. It was unclear how best to model this material, so the organosilane layer from the previous example was used. The thickness of the PAH layer might explain why *Sciacca & Monro (2014)*'s silver reflectance is peaked at $\lambda_{max} \approx 425$ nm; whereas, our simulation and other reported silver nanoparticle films (*Hutter & Fendler, 2002*) exhibit maxima at $\lambda_{max} \approx 405$ nm.

this system, the sensor's response to a mid-sized protein like NeutrAvidin (60 kDa) was simulated[8]. To our knowledge, this is the first attempt at modeling a multiplexed dip sensor containing two nanoparticle species.

A dip sensor was configured in PAME, composed of a 2.5 nm thick layer of organosilanes[9] and a 92 nm layer of mixed protein-coated nanoparticles in water. A 3-layer composite material model was used to represent the mixed nanoparticles. Materials 1 and 2 were set to 80 nm AuNPs and 60 nm AgNPs, respectively, using the dielectric functions described in Fig. 4; material 3 was set to water. Au–Au effects and Ag–Ag effects are taken into account, but PAME's 2-phase EMTs cannot account for Au–Ag effects (3-phase and N-phase EMTs (*Luo, 1997*; *Zhdanov, 2008*) will be implemented in upcoming releases). Therefore, the combined layer, $\tilde{n}_{AuAg}$, is weighted in proportion to the fill fraction as, $\tilde{n}_{AuAg} = \phi_1 \tilde{n}_{Au} + \phi_2 \tilde{n}_{Ag}$. Nanoparticle coverage was chosen so as to produce a large reflectance, with approximately equal contributions from gold and silver; the actual coverage used in *Sciacca & Monro (2014)* is not stated. Ultimately, 45.56% of the surface sites were covered in gold, and 18.64% in silver. NeutrAvidin was modeled as a 6 nm sphere (*Tsortos et al., 2008*) of dispersive refractive index, $n \approx 1.5$ (*Sarid & Challener, 2010*), filling a 6 nm-wide shell on the nanoparticles from 0 to 95% coverage (617 proteins per AuNP and 363 per AgNP), as shown in Fig. 7.

Despite using several approximations, the simulation provides many insights into multiplexed sensors. First, the 60 nm AgNPs reflect much more efficiently, despite AgNPs

and AuNPs having nearly identical extinction cross sections (Fig. 4). This is because silver particles are more efficient scatterers (*Lee & El-Sayed, 2006*), and reflectance depends exponentially on scattering cross section (*Quinten, 2011*). Therefore, reflectance sensors composed of highly-scattering particles can utilize sparse nanoparticle films, which are less susceptible to aggregation (*Scarpettini & Bragas, 2010*) and electrostatic and avidity effects. Secondly, the normalized response to protein binding is about 0.08 units for silver, and 0.06 for gold; however, there are 2.33 more proteins on gold than silver. Therefore, considering the response per molecule, 60 nm silver spheres are 3.12 time more sensitive to protein binding than 80 nm gold spheres. Experiments have confirmed similar three-fold enhancement to protein-induced plasmon resonance shifts in aqueous solutions of AuNPs and AgNPs (*Sun & Xia, 2002*; *Mayer, Hafner & Antigen, 2011*). This suggests a correspondence between shifts measured in free solution and the reflectance in optical fibers, despite little similarity in the qualitative profile of the response. *Nusz et al. (2009)* has suggested a figure of merit to objectively compare shifts vs. intensity responses.

Finally, if response is partitioned into two spectral regions, such that $385\ nm < \lambda \le 500\ nm$ corresponds to silver, and $\lambda \ge 500\ nm$ to gold, then Fig. 7 illuminates an important result: despite a clear separation in the peaks of the reflectance spectra, the response to NeutrAvidin spans both partitions. For example, in Fig. 7C the gold region ($\lambda \ge 550\ nm$) clearly responds to proteins binding to silver nanoparticles. This could lead to misinterpretations; for example, the response at $\lambda \approx 570\ nm$ could be misattributed to non-specific binding onto gold, when in fact there is only binding to silver. In *Sciacca & Monro (2014)*, both the gold and silver spectral regions responded to apoE, when only gold is coated with anti-apoE. While the signal in the silver region could be due to non-specific interactions between the anti-apoE and AgNPs, these simulations show that it could simply be due to spectral overlap in the gold and silver response, the extent of which depends on the dielectric function of the protein, the Au–Ag coupling and other factors.

## IMPLEMENTATION AND PERFORMANCE

PAME's graphical interface and event-handling framework is built on the *Enthought* Tool Suite, especially Traits and TraitsUI. Traits is particularly useful for rapid application development (*Varoquaux, 2010*). TraitsUI leverages either PyQT, Pyside or WxPython on the backend to generate the graphical interface. Some discrepancies in the user interface may be encountered between different backends, and possibly between operating systems. PAME has been tested on Ubuntu, OSX and Windows 7. A future refactor to supplant TraitsUI with Enaml (*NucleicDevelopmentTeam, 2013*) should resolve view inconsistencies.

To enhance speed, PAME utilizes Numpy (*Oliphant, 2007*) and Pandas to vectorize most of its computations. Complex structures such as multilayers of 20 or more materials, with over a thousand datapoints per sample, are reasonably handled on a low-end laptop (Intel$^{TM}$ Core 2 Duo, 4GB DDR2 RAM). The intended operating conditions for PAME are stacks of less than 10 layers, and dispersive media of 100 or fewer datapoints. At present, the main performance bottleneck is redundant event triggering. Because PAME is highly interactive, changing a global parameter such as the working spectral range will trigger

updates in every material in the multilayer stack. For nanoparticles, this means the core, medium and possibly shell materials are all recomputed, each of which triggers a separate recalculation and redraw of the Mie-scattering cross sections. Streamlining global event handlers should yield appreciable performance gains, followed by additional vectorization of the TMM calculation, and finally implementing calculations that cannot be vectorized in Cython (*Behnel et al., 2010*).

## FUTURE IMPROVEMENTS

Currently, PAME's nanoparticle support is limited to nanospheres and core–shell particles because analytical solutions to these systems exist, and because many effective medium approximations are implemented with spheres in mind. The electromagnetic properties of nanoparticles of arbitrary morphology can be solved with numerical methods such as DDA, FDTD, or BEM, and libraries like MNPBEM implement common particle morphologies out-of-the-box. The recently released PyGBe (*Cooper, Bardhan & Barba, 2014*) library brings this potential to Python. PyGBe has been used to simulate protein interactions near the surface of materials (*Lin, Liu & Wang, 2015*), meaning it has the potential to supplant the current geometrical fill models used to describe protein-nanoparticle interactions. By analyzing interactions in the near-field with PyGBe and in the far-field with PAME, comprehensive insights into nanoparticle systems may be obtained.

Even if exotic nanoparticle are incorporated into PAME, they would still need to be homogenized through an effective mixing theory to fit the TMM multilayer model. While classical EMTs can account for non-spherical inclusions through a dipole polarizability parameter (*Garcıa, Llopis & Paje, 1999*; *Quinten, 2011*), modern two-material EMTs derived from spectral density theory (*Bergman, 1978*; *Sancho-Parramon, 2011*; *Lans, 2013*), N-material generalized tensor formulations (*Habashy & Abubakar, 2007*; *Zhdanov, 2008*), and multipole treatments (*Malasi, Kalyanaraman & Garcia, 2014*) give better descriptions of real films topologies, and will be incorporated into PAME in the near future.

Some systems cannot be adequately described with EMTs, for example films composed of large, highly-scattering particles (*Quinten, 2011*). In such cases, is still possible to compute the reflectance of a film of a few hundred spheres through generalized Mie theory, which is a coherent superposition of the multipole moments of each particle, or for larger films using incoherent superposition methods (*Elias & Elias, 2002*; *Quinten, 2011*), but such approaches don't readily interface to the multilayer model. Rigorous coupled-wave analysis (RCWA) may be a viable alternative, as it as it incorporates periodic dielectric structures (*Moharam et al., 1995*) directly into TMM calculations, and is already implemented in Python (*Rathgen, 2008*; *Francis, 2014*). RCWA could be integrated into PAME without major refactoring, and has already been demonstrated as a viable alternative to EMTs in describing nanoparticle-embedded films in biosensors (*Wu & Wang, 2009*).

## CONCLUSION

Plasmonic biosensing offers a promising alternative to conventional label-free protein detection techniques like enzyme-linked immunosorbent assays (ELISA) and Western blots, but dedicated software tools for the common sensor geometries are not readily

accessible. PAME fills the gap by providing an open-source tool which combines aspects of thin-film design, effective medium theories, and nanoscience to provide a modeling environment for biosensing. In this work, it has been shown that PAME can simulate a refractometer made from a dip sensor of AuNPs, and experimental data shows good agreement without invoking extensive fit parameters. Furthermore, PAME is flexible enough to reproduce results on new multiplexed sensor designs like those proposed by *Lin et al. (2012)* and *Sciacca & Monro (2014)*. As plasmonic biosensors continue to develop, PAME should prove a useful tool for characterizing sensor response, a necessary step towards *in-situ* studies.

## ABOUT

PAME documentation, source code, examples and video tutorials are hosted at: https:// github.com/hugadams/PAME. We are looking for developers to help extend the project. Please contact if interested.

**Programming Language:** Python 2.7

**License:** 3-Clause BSD

**Version:** 0.3.2

**Dependencies:** Enthought Tool Suite, Pandas, scipy (IPython $\geq$ 2.0 or greater and scikit-spectra recommended)

**OS:** Windows, Mac and Linux

**Persistent Identifier:** DOI 10.5281/zenodo.17578

**Binary Installers:** Under development

## ACKNOWLEDGEMENTS

We'd like to thank Robert Kern and Jonathan March for many helpful discussions on Traits and TraitsUI, and Rayhaan Rasheed for helping to create the illustrations.

### Funding

This work was supported in part by the George Gamow Research Fellowship, Luther Rice Collaborative Research fellowship programs, the George Washington University (GWU) Knox fellowship and the GWU Presidential Merit Fellowship. The funders had no role in study design, data collection and analysis, decision to publish, or preparation of the manuscript.

### Grant Disclosures

The following grant information was disclosed by the authors:
George Gamow Research Fellowship.
Luther Rice Collaborative Research Fellowship.
George Washington University Knox Fellowship.
GWU Presidential Merit Fellowship.

## Competing Interests

The authors declare there are no competing interests.

## Author Contributions

- Adam Hughes analyzed the data, contributed reagents/materials/analysis tools, wrote the paper, prepared figures and/or tables, performed the computation work.
- Zhaowen Liu analyzed the data, contributed reagents/materials/analysis tools, prepared figures and/or tables, performed the computation work.
- Mark E. Reeves wrote the paper, reviewed drafts of the paper.

## Data Availability

The following information was supplied regarding the deposition of related data:

Source code is hosted on github:

https://github.com/hugadams/PAME

PAME v. 0.3.2 is archived on Zenodo

DOI 10.5281/zenodo.17578

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
