# Peer review of "PAME: plasmonic assay modeling environment"

_PeerJ Computer Science, doi:10.7717/peerj-cs.17_

## Round 0.1 · original submission · Minor Revisions

Overall the paper is well organized and presented since a big portion of it has been posted online. Authors should revise the paper according to the comments of the reviewers.

Reviewer 1 ·

Basic reporting

In the abstract, the author stated that the plasmonic assays own the advantages such as labeling free and measurement in real time. The combination of nanoparticles and fiberoptics show promise in in-situ probing and implantable devices. But the author did not provide information to support those advantages in the article. It will be better if the advantages of plasmonic assays and combination of nanoparticles and fiberoptics can be moved to the "Introduction" Chapter.

The abstract should focus on building, verify the model with PAME platform.

Experimental design

In Figure 4, the horizontal coordinate seems represent the wave length. What is the meaning of “counts” in vertical coordinate? How the normalized counts in Figure 6c and 6d were obtained? Provide detail information will make those clearer.

Validity of the findings

No Comments.

·

Basic reporting

This article meet all standards of PeerJ with a very high quality.
For example, the structure of the article are well organized, content are self-sustained with appropriate introduction and background information. Data and source are openly shared online.

Experimental design

Several experimental examples are given in this article to show how to use the open-sourced tool PAME.
Experiments are clear described and results are well discussed. A very high technical standard are maintained in the experiment design.
Meanwhile, supplemental examples are also abundant and very helpful for new users.

Validity of the findings

The software introduced in this article shows a very convenient interface, comprehensive functionalities for modeling plasmonic systems, as well as many previous modeling examples in literature.

With the authors open-sourcing the software, it will be easily accessible to researchers in the area, and bring a great contribution to the community.

Additional comments

The submitted paper discuss the plasmonic assay modeling and simulation, and introduced an open-sourcePython application for modeling plasmic systems of bulk and nanoparticle-embedded metallic films.

The article is of very high quality in terms of basic structure, scientific reporting, theory and experiment design, and final validation.

Very minor issues are: 1. a few ? marks in Page 1( in Introduction), and Page 11 (2 paragraph before section 4 IMPLEMENTATION AND PERFORMANCE).
2. Format issue, Page 2, COMSOL Multiphysics is extending to margin area.

Reviewer 3 ·

Basic reporting

Please give more introduction and analysis for the section 4 implementation and performance

Experimental design

The experimental design is well organized and reasonable

Validity of the findings

The data can support authors’ ideas and conclusions

Additional comments

This paper has innovated idea and good simulation; it is also well organized. If the authors can provide more introduction and analysis for section 4 implementation and performance, it is much better.

---

## Round 0.2 · accepted · Accept

The paper is accepted. Please do the final minor edit as noted by Reviewer 4.

Reviewer 1 ·

Basic reporting

The author introduced open-source Python application for modeling plasmonic systems of bulk and nanoparticle-embedded metallic films, and also present the PAME’s theory and design. The paper was well organized and the examples are suitable.

Experimental design

After building the model, the author gave two examples for the application, one is Refractometer and another on e is Multiplexed Ag-Au Sensor. Those two examples are sufficiently support the author's ideas.

Validity of the findings

The examples in this paper are simulations of a fiberoptic refractometer,and protein binding to a multiplexed sensor composed of a mixed layer of gold and silver colloids.
The examples, data and citation can support the author's view points in modeling the plasmonic system.

Additional comments

The author introduced a method to model the plasmonic systems based on open-source Python, and also provide some examples and data to support the view points. This modeling method is meaningful in analyze the plasmonic systems.

.

·

Basic reporting

This article meet all standards of PeerJ with a very high quality.

Experimental design

Experimental design are carefully described and results are well discussed.

Validity of the findings

No additional comments.

Additional comments

Again, this article is of very high quality in terms of basic structure, scientific reporting, theory and experiment design. The authors had fixed previous issues and now is fully ready for publication.

Reviewer 3 ·

Basic reporting

No comments

Experimental design

No comments

Validity of the findings

No comments

·

Basic reporting

The authors reported a python-based software platform that could model and simulate the behavior of plasmonic assays assuming certain experimental conditions. PAGE also provides an user friendly interface for modeling and simulation.

Experimental design

The authors reported three examples of how PAGE could be utilized. In the first example, PAGE was used to simulate the sensor response to increasing concentrations of glycerine for an axial fiber. It was shown that there was a strong agreement of between measured and PAGE simulated response. In the second example, PAGE was used to model a multiplexed dip sensor containing gold and silver nanoparticle species. Despite several approximations, PAGE was able to provide several insights into the multiplexed sensor.

Validity of the findings

In the first example, PAGE’s simulated results aligned well with the experimental results. In the second example, PAGE’s simulated results were supported by several lines of evidence from previous publications.

Additional comments

Page 2, change “PAME if a fully graphical application that …” to “PAME is a fully graphical application that …”